# Token Merging for Training-Free Semantic Binding in Text-to-Image Synthesis

**Taihang Hu**[1], **Linxuan Li**[1], **Joost van de Weijer**[3], **Hongcheng Gao**[4]
**Fahad Shahbaz Khan**[5,6], **Jian Yang**[1], **Ming-Ming Cheng**[1,2], **Kai Wang**[3*], **Yaxing Wang**[1,2*]

[1]VCIP, College of Computer Science, Nankai University, [2]NKIARI, Shenzhen Futian
[3]Computer Vision Center, Universitat Autònoma de Barcelona
[4]University of Chinese Academy of Sciences
[5]Mohamed bin Zayed University of AI, [6]Linkoping University
{hutaihang00, linxuanli520, gaohongcheng2000}@gmail.com
{joost, kwang}@cvc.uab.es, fahad.khan@liu.se
{csjyang,cmm,yaxing}@nankai.edu.cn

## Abstract

Although text-to-image (T2I) models exhibit remarkable generation capabilities, they frequently fail to accurately bind semantically related objects or attributes in the input prompts; a challenge termed *semantic binding*. Previous approaches either involve intensive fine-tuning of the entire T2I model or require users or large language models to specify generation layouts, adding complexity. In this paper, we define semantic binding as the task of associating a given object with its attribute, termed *attribute binding*, or linking it to other related sub-objects, referred to as *object binding*. We introduce a novel method called *Token Merging* (*ToMe*), which enhances semantic binding by aggregating relevant tokens into a single *composite token*. This ensures that the object, its attributes and sub-objects all share the same cross-attention map. Additionally, to address potential confusion among main objects with complex textual prompts, we propose *end token substitution* as a complementary strategy. To further refine our approach in the initial stages of T2I generation, where layouts are determined, we incorporate two auxiliary losses, an entropy loss and a semantic binding loss, to iteratively update the composite token to improve the generation integrity. We conducted extensive experiments to validate the effectiveness of *ToMe*, comparing it against various existing methods on the T2I-CompBench and our proposed GPT-4o object binding benchmark. Our method is particularly effective in complex scenarios that involve multiple objects and attributes, which previous methods often fail to address. The code will be publicly available at https://github.com/hutaihang/ToMe.

## 1 Introduction

Text-to-image generation has seen significant advancements with the recent introduction of diffusion models [57, 59, 62], with their capabilities of generating high-fidelity images from text prompts. Despite these achievements, aligning the generated images with the text prompts, which is referred to as *semantic alignment* [30, 43], remains a notable challenge. One of the most common issues observed in existing text-to-image (T2I) generation models is the lack of proper *semantic binding*, where a given object is not properly binding to its attributes or related objects. For example, as illustrated in Fig. 1, even a state-of-the-art T2I model such as SDXL [53] can struggle to generate content that accurately reflects the intended nuances of text prompts. To address the persistent challenges of aligning T2I diffusion models with the intricate semantics of text prompts, a variety of enhancement

---

[*]: Co-corresponding authors

38th Conference on Neural Information Processing Systems (NeurIPS 2024).

a **dog wearing hat** and a **cat wearing sunglasses**     a **dog with blue bow tie** and a **cat with red scarf**

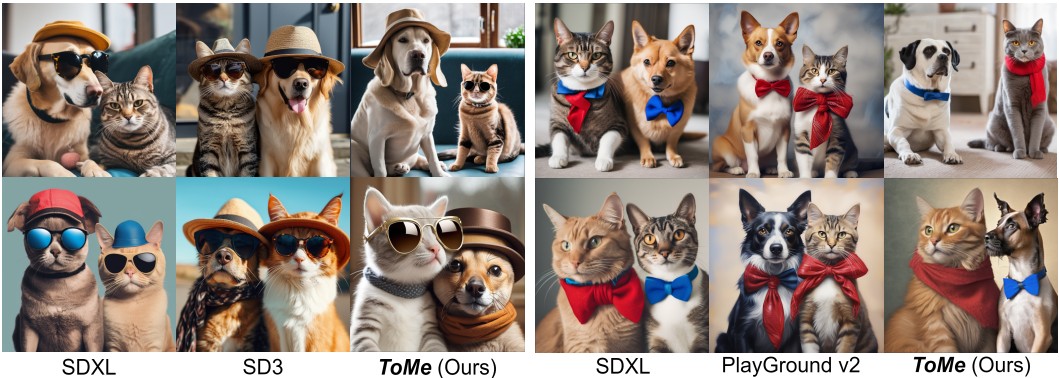

SDXL         SD3         *ToMe* (Ours)          SDXL       PlayGround v2   *ToMe* (Ours)

Figure 1: Current state-of-the-art T2I models often struggle with semantic binding in generated images according to textual prompts. For example, hats and sunglasses are placed on incorrect objects. We introduce a novel method *ToMe* to address these challenges.

strategies [35, 46, 87] are proposed, either by optimizing the latent representations [69, 82, 83], guiding the generation by layout priors [54, 71, 85] or fine-tuning the T2I models [21, 34]. Despite these advancements, these methods still encounter limitations, particularly in generating high-fidelity images involving complex scenarios where an object is binding with multiple objects or attributes.

In this paper, we categorize *semantic binding* into two categories. First, *attribute binding* involves correctly associating objects with their attributes, a topic that has been studied in prior work [58]. Second, *object binding*, which entails effectively linking objects to their related *sub-objects* (for example, a 'hat' and 'glasses'), is less explored in the existing literature. Previous methods often struggled to address this aspect of semantic binding. One of the main problems is the misalignment of objects with their corresponding sub-objects. Existing solutions address this through an explicit alignment process of the attention maps [7, 43] or by factorizing the generation projects into layout phases and generation phase [55]. In this paper, we propose a simple solution to the attention alignment problem called *token merging* (*ToMe*). Instead of multiple attention maps, which can be misaligned, we join these objects in a single *composite token* that represents the object and its attributes and sub-objects. This composite token has a single cross-attention map that ensures semantic alignment. The composite token is simply constructed by summing the CLIP text embeddings of the various tokens it represents. For example, the phrase "a dog with hat" is abbreviated as "a dog*" by aggregating the text embeddings corresponding to the last three words, as shown in Fig. 4. To justify the applied embedding addition in *ToMe*, we experimented with the semantic additivity of the text embeddings (in Fig. 3). Furthermore, to mitigate potential semantic misalignment in the end tokens from the long sequences, we propose *end token substitution* (ETS) technique.

As the T2I generation predominantly determines the layout during earlier phases [27], we introduce an entropy loss and a semantic binding loss to update the token embeddings in early steps, integrating *ToMe* with an iterative update for the composite tokens. The entropy loss is defined as the entropy of the cross-attention map corresponding to the updated composite token. This loss aims to enhance generation integrity by ensuring diverse attention across relevant areas of the image, thereby preventing focusing on non-essential regions. The semantic binding loss encourages the new learned token to infer the same noise prediction as the original corresponding phrase. This alignment further reinforces the semantic coherence between the text and the generated image.

Our final method *ToMe* is quantitatively assessed using the widely adopted T2I-CompBench [31] and our proposed GPT-4o [1] *object binding* benchmark. Comparative evaluations against various types of approaches reveal that *ToMe* outperforms them by a significant margin. Remarkably, our approach is user-friendly, requiring no dependence on large language models or specific layout information. In qualitative evaluations, we notably achieve superior generation quality, particularly in scenarios involving multi-object multi-attribute generation. This further underscores the superiority of our method. In summary, the main contributions of this paper are as follows:

- We analyze the problem of semantic binding, and highlight the role of the [EOT] token (Fig. 2), and the problems with misaligned cross-attention maps (Fig. 7). In addition, we explore token additivity as a possible solution (Fig. 3).

- We introduce a *training-free* approach called *Token Merging* (Fig. 4), denoted as *ToMe*, as a more efficient and robust solution for semantic binding. It is further enhanced by our proposed *end token substitution* and iterative *composite token* updates techniques.

- In experiments conducted on the widely used T2I-CompBench benchmark and our GPT-4o object binding benchmark, we compared *ToMe* with various state-of-the-art approaches and consistently outperformed them by significant margins.

## 2   Related works

A critical drawback of current text-to-image models is related to their limited ability to faithfully represent the precise semantics of input prompts, commonly referred to as *semantic alignment*. Various studies have identified common semantic failures and proposed mitigation strategies. They can be roughly categorized into four main streams.

**Optimization-based methods** primarily adjust text embeddings [20, 65] or optimize noisy signals to strengthen attention maps [26, 48, 63, 69, 82, 83]. These methods are basically inspired by the observations from text-based image editing methods [27, 40, 64, 66], suggesting that the layouts of objects are determined by self-attention and cross-attention maps from the UNet of the T2I diffusion models. For example, Attend-and-Excite [7] improves object existence by exciting the attention score of each object. Divide-and-Bind [43] improves by maximizing the total variation of the attention map to prompt multiple spatially distinct attention excitations. SynGen [58] syntactically analyzes the prompt to identify entities and their modifiers, and then uses attention loss functions to encourage the cross-attention maps to agree with the linguistic binding reflected in the syntax. A-star [2] proposes to minimize concept overlap and change in attention maps through iterations. Composable Diffusion [45] decomposes complex texts into simpler segments and then composes the image from these segments. Structure Diffusion [20] attempts to address this by leveraging linguistic structures to guide the cross-attention maps. Rich-Text [24] enriches textual prompts by incorporating various formatting controls and decomposes the generation task into merging inferences from multiple region-based diffusions. However, these methods often fail in complex scenarios that generate multiple objects or multiple attributes.

**Layout-to-Image methods** [4, 9, 14, 17, 25, 32, 36, 47] are widely using layouts, particularly in the form of bounding boxes or segmentation maps, as a popular intermediary to bridge the gap between text input and the generated images. For example, BoxDiff [73] encourages the desired objects to appear in the specified region by calculating losses based on the maximum values in cross-attention maps. Similarly, Attention-Refocusing [52] modifies both cross-attention and self-attention maps to control object positions. BoxNet [67] first trains a network to predict the box for each entity that possesses the attribute specified in the prompt, and then force the generation to follow the attention mask control. Additionally, InstanceDiffusion [68] enhances text-to-image models by providing extra instance-level control. There are also finetuning methods [5, 42, 50, 79] allow for additional layout conditions after fine-tuning over pair images, which are not specifically designed to solve the *semantic alignment* problem. Despite their promise, these methods obviously prolong the training time. Furthermore, the application of layout priors is challenging when it comes to global background descriptions or abstract elements. This limitation constrains the versatility of these techniques, making it difficult to deploy them effectively across real scenarios where non-specific spatial arrangements are crucial.

**LLM-augmented methods** are mainly following text-to-layout-to-image generation pipelines [15, 23, 33, 44, 55, 65, 80, 81, 86], first to generate layouts from large language models (LLMs) and force the T2I generations to follow this guidance as the previous layout-guided methods. Some methods, such as RPG [75] and MuLan [39], harness the powerful chain of thought reasoning ability of multimodal LLMs to enhance the compositionality of text-to-image diffusion models.

**Finetuning-based methods** [13, 76] update the model parameters over huge datasets to augment the semantic alignment. Among them, CoMat [34] proposes an end-to-end fine-tuning strategy for text-to-image diffusion models by incorporating image-to-text concept matching. ELLA [30] equips text-to-image diffusion models with powerful Large Language Models (LLM) to enhance text alignment by bridging these two pre-trained models with trainable semantic alignment connectors. More recently, Ranni [21] improves T2I generation by bridging the text and image with a semantic panel with LLMs and is fine-tuned over an automatically prepared semantic panel dataset. There

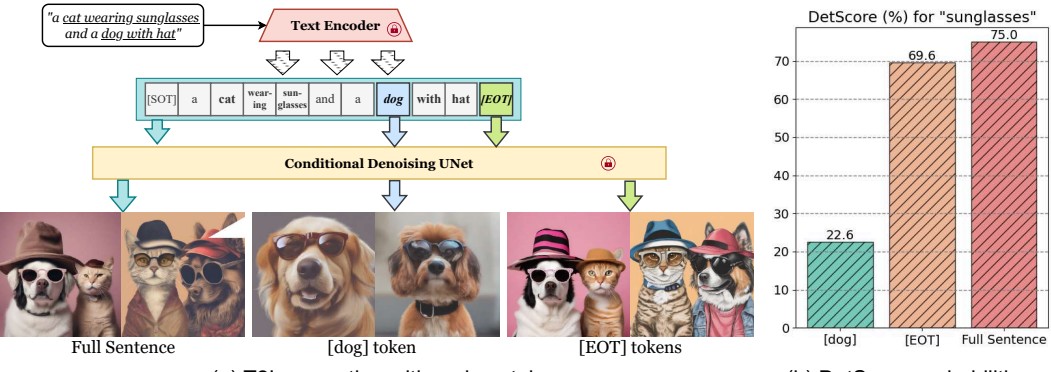

| (a) T2I generation with various tokens | (b) DetScore probabilities |

Figure 2: We generate images with various input prompts in (a): "a cat wearing sunglasses and a dog wearing a hat"; the single-token embedding [dog]; the end token [EOT] . (b) After that, we compute the probability of containing "sunglasses" in the generated images in subfigure .

are also improved T2I models [10, 11, 51] learning from scratch over huge datasets. These methods improve semantic alignment implicitly by better architecture design and larger amount of training data. They further demand marvelous computational resources to achieve the purpose.

In this paper, we tackle the *semantic binding* problem, which is a broad subcase of *semantic alignment*, in a training-free manner, neither needing the LLMs nor any training over additional datasets. Furthermore, we achieve better performance when facing complex T2I generation scenarios where users require multiple objects or multiple attributes related to a specific object.

## 3 Methods

Semantic binding in T2I generation refers to the crucial requirement of establishing accurate associations between objects and their relevant attributes or related sub-objects. This process avoids semantic misalignment in the generated images, ensuring that each visual element aligns correctly with its descriptive cues in the text. In this section, we begin by providing the preliminaries. Subsequently, we illustrate the motivation through a series of experimental analyses (Sec. 3.1). Finally, we elaborate on our methods in detail (Sec. 3.2). An illustration of our method *ToMe* is shown in Fig. 4.

**Latent Diffusion Models.** We build our novel approach for semantic alignment on the standard SDXL [53] model. The model is composed of two main parts: an autoencoder (i.e., a encoder $\mathcal{E}$ and a decoder $\mathcal{D}$ ) and a diffusion model (i.e., $\epsilon_\theta$ with parameter $\theta$). The model $\epsilon_\theta$ is updated by the loss:

$$L_{LDM} := \mathbb{E}_{z_0 \sim \mathcal{E}(x), y, \epsilon \sim \mathcal{N}(0,1), t \sim \text{Uniform}(1,T)} \left[ \| \epsilon - \epsilon_\theta(z_t, t, \tau_\xi(\mathcal{P})) \|_2^2 \right], \quad (1)$$

where $\epsilon_\theta$ is a UNet, conditioning a latent input $z_t$, a text embedding $\tau_\xi(\mathcal{P})$ and a timestep $t \sim$ Uniform$(1, T)$. More specifically, text-guided diffusion models aim to generate an image from random noise $z_T$ and a conditional input prompt $\mathcal{P}$. To distinguish from the general conditions in LDMs, we itemize the textual condition as $\mathcal{C} = \tau_\xi(\mathcal{P})$, where $\tau_\xi$ is the CLIP text encoder [56][†]. The cross-attention map is obtained from $\epsilon_\theta(z_t, t, \mathcal{C})$. Let $f_{z_t}$ be a feature map output of the network $\epsilon_\theta$. We get a query matrix $Q_t = l_Q(f_{z_t})$ with projection network $l_Q$. Similarly, given a textual embedding $\mathcal{C}$, we compute a key matrix $\mathcal{K} = l_\mathcal{K}(\mathcal{C})$ with projection network $l_\mathcal{K}$. Then the attention map is computed according to: $\mathcal{A}_t = softmax(Q_t \cdot \mathcal{K}^T / \sqrt{d})$ where $d$ is the latent dimension, and the cell $[\mathcal{A}_t]_{ij}$ defines the weight of the $j$-th token on the $i$-th token.

### 3.1 Text Embedding Analysis

To address the semantic binding problem, we concentrate on the text embeddings utilized during the diffusion model generation process, as they predominantly dictate the content of the generated images. For a given text prompt $\mathcal{P}$, it is tokenized by the CLIP text model by padding a start

---

[†]SDXL uses two CLIP text encoders and concatenate the two text embeddings as the final text embedding.

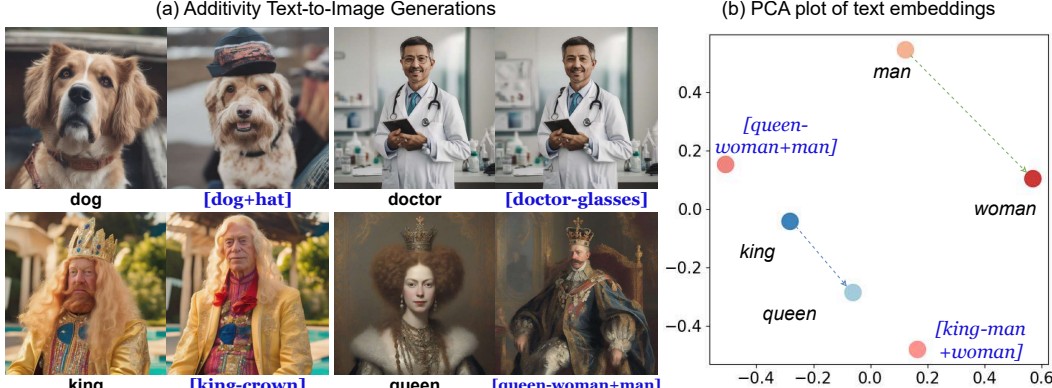

Figure 3: (a) Image generations with the property of token additivity. All images are generated by the prompt template "a photo of a {*object*}." (b) PCA plot for additivity of text embeddings.

token [SOT] and several end tokens [EOT] to extend its length to $M$(=77 by default). After the CLIP text encoder $\tau_\xi$, the condition is formulated as $\mathcal{C} = \tau_\xi(\mathcal{P})$. Each row in $\mathcal{C}$ represents a corresponding token embedding after the CLIP text transformers. For example, the text embedding for the sentence $\mathcal{P}$ ="a cat wearing sunglasses and a dog wearing a hat" is represented as: $\mathcal{C} = [\boldsymbol{c}_0^{SOT}, \boldsymbol{c}_1^a, \boldsymbol{c}_2^{cat}, \cdots, \boldsymbol{c}_7^{dog}, \boldsymbol{c}_8^{wearing}, \boldsymbol{c}_9^{hat}, \boldsymbol{c}_{10}^{EOT}, \cdots, \boldsymbol{c}_{M-1}^{EOT}]$. In the following analysis, we take this as a default example (except when defined differently).

**Information Coupling.** We begin by generating images conditioning on the textual embedding $\mathcal{C}$, as illustrated in the first two columns at the bottom of Fig. 2-(a). We observe that the attributes appear in a misalignment between the dog and the cat. Subsequently, we extract the token embedding $\boldsymbol{c}_7^{dog}$ from the textual embedding and input it to the UNet $\epsilon_\theta$ (i.e., $\mathcal{C} = [\boldsymbol{c}_7^{dog}]$)[‡]. As depicted in the middle columns of Fig. 2-(a). The dog object is frequently wearing glasses, further highlighting the semantic leakage issue. Furthermore, when we take $\mathcal{C}^{[EOT]} = [\boldsymbol{c}_{10}^{EOT}, \cdots, \boldsymbol{c}_{M-1}^{EOT}]$ as input, the generated images closely resemble all information obtained using the entire textual embedding $\mathcal{C}$. As the [EOT] interacts with all tokens, it often encapsulates the entire semantic information [41, 72].We further report the *DetScore* [12] to show the probability of containing the corresponding object ("sunglasses") in the generated 100 images. As illustrated in Fig. 2-(b), for these three different cases, the DetScore is 22.6%, 69.6% and 75.0%, respectively. These findings also align with our observations above.

**Additivity Property.** Inspired by the semantic additivity of the text embeddings in previous research[6, 49], we experiment the additive property of the CLIP textual embedding. We represent the textual embedding corresponding to the prompt "a photo of a dog" as $\mathcal{C}_1 = [c_0^{SOT}, c_1^a, \cdots, c_5^{dog}, c_6^{EOT}, \cdots, c_{M-1}^{SOT}]$. The textual embedding for the prompt "a photo of a hat" is represented as $\mathcal{C}_2 = [c_0^{SOT}, c_1^a, \cdots, c_5^{hat}, c_6^{EOT}, \cdots, c_{M-1}^{EOT}]$. Next, we perform element-wise addition between the object tokens (i.e., $c_5^{dog}$ and $c_5^{hat}$) and the corresponding [EOT] tokens. Specifically, the resulting new embedding is $\mathcal{C}' = \textbf{Concat}(\mathcal{C}_1[0:4], \mathcal{C}_1[5:M-1] + \mathcal{C}_2[5:M-1])$. Afterward, the textual embeddings $\mathcal{C}'$ are input into the diffusion UNet to generate the images shown in Fig. 3-(a). We can observe that this additivity property allows adding objects (up-left), removing objects (up-right, down-left) and even complex semantic computations (down-right). To explore the mechanism behind this phenomenon, we conducted PCA dimensionality reduction visualization on the token representations of each prompt, as illustrated in Fig. 3-(b). The directional vector obtained from "queen-king" is approximately identical to that of "woman-man" with the cosine similarity of 0.998.

**In conclusion**, our analysis shows that the semantic content of text tokens is coupled and entangled, resulting in attribute confusion across different subjects. Moreover, we found that in diffusion models, text embeddings exhibit semantically additive properties. This implies that the diffusion model is capable of interpreting a composite token, derived from the summation of multiple individual tokens, integrating the semantic attributes of the combined tokens.

---

[‡]Note in this case, the size of the input textual embedding is $1 \times 2048$ instead $77 \times 2048$.

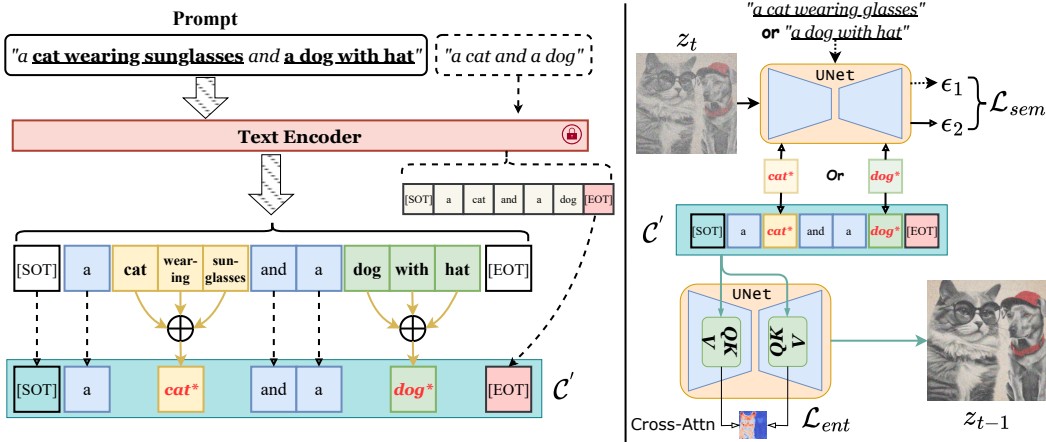

(a) Token Merging and End Token Substitution

(b) Iterative Composite Token Update

Figure 4: *ToMe* is composed of two parts: one with Token Merging and end token substitution, and the other token updating part with two auxiliary losses for iterative *composite token* update.

## 3.2 *ToMe*: Token Merging

Suppose the initial prompt $\mathcal{P}$ contains $K$ entities indicated by noun words and their corresponding tokens as $\{n^1, ..., n^k..., n^K\}$. Each entity is often related to a token with relevant objects or attributes set as $(n^k, a^k)$. For example, in the sentence "a cat wearing glasses and a dog with a hat", $n^1 =$ <cat>, $a^1 = \{$<wearing>,<glasses>$\}, n^2 =$ <dog>, $a^2 = \{$<with>,<a>,<hat>$\}$.

### 3.2.1 Token Merging techniques

The semantic additivity of token embeddings inspires us to achieve co-expression of entities and attributes by explicitly binding tokens together. We employ element-wise addition to accomplish semantic merging of tokens. For a prompt $\mathcal{P}$ containing $K$ entities, we fuse each subject-attribute pair $(n^k, a^k)$ into $\hat{c}_k = n^k + \sum a^k$, referred to as a *composite token*. This innovative approach introduces an additional benefit by utilizing a single composite token to condense a lengthy prompt sequence, resulting in a unified cross-attention map, thus avoid semantic misalignment. Such observations are further shown in the ablation study and appendix.

**End Token Substitution (ETS).** Meanwhile, as the semantic information contained in [EOT] can interfere with attribute expression, we mitigate this interference by replacing [EOT] to eliminate attribute information contained within them, retaining only the semantic information of each subject. For instance, when the prompt is "a cat wearing hat and a dog wearing sunglasses," we use the [EOT] obtained from the prompt "a cat and a dog" to replace the original [EOT] . As illustrated in Fig. 4-a, the final text embedding after subject-attribute enhancement and EOT replacement is $\mathcal{C} = \left[ \boldsymbol{c}_0^{SOT}, \boldsymbol{c}_1^a, \boldsymbol{c}_2^{dog*}, \cdots, \boldsymbol{c}_5^{cat*}, \boldsymbol{c}_6^{EOT*}, \cdots, \boldsymbol{c}_{76}^{EOT*} \right]$. Here, dog* and EOT* respectively denote tokens after token merging and end token substitution.

### 3.2.2 Iterative composite Token Update

**Semantic binding loss.** As stated in section 3.1, the semantic information of each token embedding is inherently linked. After strengthening the relationship between subjects and their attributes, it becomes crucial to eliminate any irrelevant semantic information within the composite tokens to prevent misrepresentation of attributes. As illustrated in Fig. 4-(b), to ensure that the semantics of the composite tokens correspond accurately to the noun phrases they are meant to represent, we employ a clean prompt as a supervisory signal. Specifically, for a composite token embedding $\hat{c}^{dog}$, which corresponds to the noun phrase "a dog wearing hat", we aim for the diffusion model to exhibit consistent noise prediction for this composite token and the full phrase. In mathematical terms, this objective can be expressed as ensuring that $\epsilon_\theta(z_t, \hat{c}^{dog}, t) \approx \epsilon_\theta(z_t, \mathcal{C}, t)$. This effectively aligns $\nabla_{z_t} \log P_\theta(z_t | \hat{c}^{dog}) \approx \nabla_{z_t} \log P_\theta(z_t | \mathcal{C})$ [18, 28]. At time step $t$, we use the semantic binding loss to align token semantics $\mathcal{L}_{sem} = \sum_{k \in [1,K]} \| \epsilon_\theta(z_t, \hat{c}_k, t) - \epsilon_\theta(z_t, \mathcal{C}, t) \|_2^2$.

**Entropy loss.** Following that, we calculate the information carried by each token embedding through entropy statistics. As shown in Fig. 7, we extract the cross-attention map $\mathcal{A}_k$ corresponding to the $k$-th token[27]. After normalizing the cross-attention map as $\sum_{p_i \in \mathcal{A}_k} p_i = 1$, we compute the entropy of each token as $entropy(\mathbf{token}_k) = \sum_{p_i \in \mathcal{A}_k} -p_i \log(p_i)$. Decreasing the entropy of the cross-attention maps can help ensure that tokens focus exclusively on their designated regions, thereby preventing the cross-attention map from becoming overly divergent. This is further depicted in Fig. 7, where we observe instances of attribute confusion, characterized by different tokens inappropriately influencing the same image region. The entropy regularization loss is defined as $\mathcal{L}_{ent} = \sum_{k \in [1,K]} \sum_{p_i \in A_k} -p_i \log(p_i)$ during time step $t$.

Finally, the overall $\mathcal{L} = \mathcal{L}_{ent} + \lambda \cdot \mathcal{L}_{sem}$ is computed by these two novel losses to update the *composite token* during each time $t < T_{opt}$ and $\lambda$ is the trade-off hyperparameter.

## 4 Experiments

### 4.1 Experimental Setups

**Evaluation Benchmarks and Metrics.** We evaluate the effectiveness of *ToMe* over T2I-CompBench [31], a comprehensive benchmark for open-world compositional T2I generations, encompassing attribute binding and object relationships. We focus on the semantic binding problem, where T2I-CompBench predominantly evaluates through three attribute subsets (i.e., color, shape, and texture). We follow the evaluation protocol [21, 30, 34] that using 300 validation prompts for evaluation under each subset and the BLIP-VQA score[31] as the evaluation metrics. Following that, we adopt the ImageReward [74] model to evaluate human preference scores, which comprehensively measure image quality and prompt alignment. To comprehensively evaluate *object binding* performance, we introduce a new **GPT-4o Benchmark** of 50 prompts using the template "a [objectA] with a [itemA] and a [objectB] with a [itemB].". For example, objectA and objectB are objects like "cat" and "dog" while itemA and itemB are associated items "hat" and "glasses". Afterward, we used the multimodal model GPT-4o [1] to compute the consistency score between the generated images and the prompts for objective assessment. More details are available in the Appendix C.5.

**Implementation Details.** We used SDXL [53] as our base model. To automate image generation for evaluation, we employed SpaCy [29] for syntactic parsing of prompts to identify each object and its corresponding attributes for token merging. The iterative composite token update is performed during the first 20% of the denoising steps $T_{opt} = 0.2T$.

**Comparison Methods.** To evaluate our method's effectiveness, we compared the current state-of-the-art methods. These primarily encompass: (1) state-of-the-art T2I diffusion models, including SDXL [53], Playground-v2 [37] (2) Finetuning-based methods, including CoMat [34], ELLA [30] (3) Optimization-based method SynGen [58] (4) LLM-augmented finetuning-based method Ranni [21]. More comparison results are shown in the Appendix E.

### 4.2 Experimental Results

**Quantitative Comparison.** As shown in Table 1, *ToMe* consistently outperforms or performs comparably to existing methods in BLIP-VQA scores across the color, texture, and shape attribute binding subsets, indicating its effectiveness in avoiding attribute confusion. Human-preference scores evaluated through the ImageReward[74] model(note that the model scores are logits and can be negative) suggest that images generated by *ToMe* can better align with prompts. Specifically, despite ELLA's[30] use of LLama or T5-XL to replace the CLIP Text Encoder for stronger text embeddings, our method still achieves higher BLIP-VQA scores compared to ELLA. The significant improvement in GPT-4o scores also demonstrates the effectivenes of *ToMe* in *object binding*.

**Qualitative Comparison.** Following SynGen [58], we classify the failure cases of *attribute binding* into three main categories. (i) Semantic leak in prompt, where the attribute $a^k$ is not corresponding to its entity $n^k$; (ii) Semantic leak out of prompt, where the attribute $a^k$ is describing the background or some entity not referred to in the prompt $\mathcal{P}$; (iii) Attribute neglect, where the attribute $a^k$ is totally ignored in the image generation. Fig. 5 presents our qualitative comparison results with other methods. The first three rows show more complex *object binding* results, while the last two rows demonstrate attribute binding results. The semantic binding errors in images generated by SDXL[53]

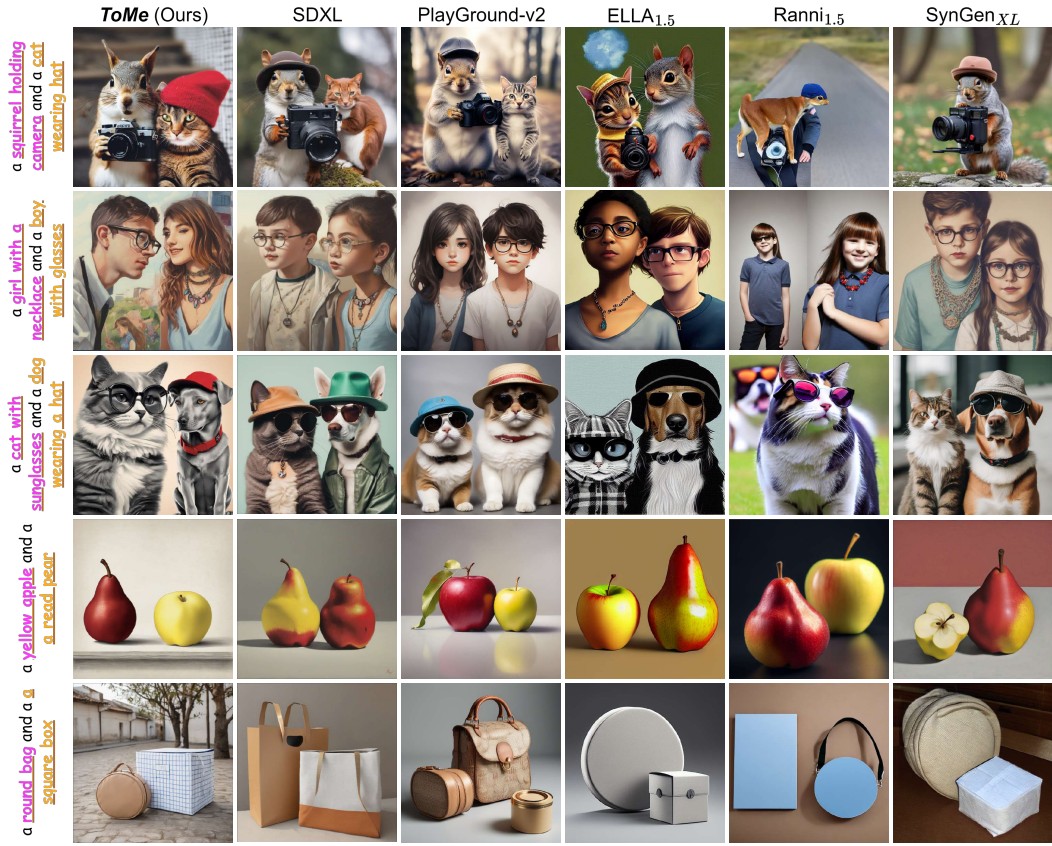

Figure 5: Qualitative comparison among various T2I generation methods with complex prompts.

can largely be attributed to (i) semantic leak in the prompt, as evidenced in the first and second row. Playground-v2[37] confronts similar semantic binding issue as SDXL. ELLA[30] can occasionally succeed in simple attribute binding as in the fifth row, but it frequently encounters (i) semantic leak in the prompt and (iii) attribute neglect errors as shown in the first three prompts. Ranni [21] generates images based on layouts created by a large language model, which can partially address more complex object binding (second row). However, layout-based methods may encounter constrains in achieving proper image layouts, such as shown in the first row with complex descriptions. SynGen [58], which focus on attribute binding problems, achieves good results in color and shape binding but fails in object binding, exhibiting varying degrees of (i) and (iii) failures. Compared to these methods, our

Table 1: Quantitative results for semantic binding assessment on various benchmarking subsets. We denote the best score in blue , and the second-best score in green .

| Method | Base Model | Train | BLIP-VQA ↑ | | | Human-preference ↑ | | | GPT-4o ↑ |
|---|---|---|---|---|---|---|---|---|---|
| | | | Color | Texture | Shape | Color | Texture | Shape | |
| SDXL[53] | - | ✓ | 0.6369 | 0.5637 | 0.5408 | 0.7798 | 0.5140 | 0.4029 | 0.4907 |
| PlayG-v2[37] | - | ✓ | 0.6208 | 0.6125 | 0.5087 | - | - | - | 0.5417 |
| Ranni[21] | SD1.5 | ✓ | 0.2414 | 0.3029 | 0.2857 | -0.8554 | -0.6853 | -0.8051 | 0.4166 |
| ELLA[30] | | ✓ | 0.6911 | 0.6308 | 0.4938 | 0.6586 | 0.2963 | 0.0565 | 0.6481 |
| SynGen[58] | | ✗ | 0.6619 | 0.6451 | 0.4661 | 0.4326 | 0.5072 | 0.0426 | 0.5545 |
| CoMat[34] | | ✓ | 0.6561 | 0.6190 | 0.4975 | - | - | - | - |
| Ranni[21] | SDXL | ✓ | 0.6893 | 0.6325 | 0.4934 | - | - | - | - |
| ELLA[30] | | ✓ | 0.7260 | 0.6686 | 0.5634 | - | - | - | - |
| SynGen[58] | | ✗ | 0.7010 | 0.6044 | 0.5069 | 1.016 | 0.7867 | 0.4016 | 0.6458 |
| CoMat[34] | | ✓ | 0.7774 | 0.6591 | 0.5262 | - | - | - | - |
| *ToMe* (Ours) | SDXL | ✗ | 0.7656 | 0.6894 | 0.6051 | 1.074 | 0.9281 | 0.5916 | 0.9549 |

Table 2: Ablation Study conducted on the T2I-CompBench benchmark.

| Conf. | ToMe | $\mathcal{L}_{ent}$ | $\mathcal{L}_{sem}$ | BLIP-VQA Color | Texture | Shape |
|---|---|---|---|---|---|---|
| A | × | × | × | 0.6369 | 0.5637 | 0.5408 |
| B | ✓ | × | × | 0.6577 | 0.5828 | 0.5437 |
| C | ✓ | ✓ | × | 0.7525 | 0.6775 | 0.5797 |
| D | × | ✓ | ✓ | 0.5881 | 0.6194 | 0.5386 |
| E | × | ✓ | × | 0.5983 | 0.5798 | 0.5125 |
| F | ✓ | × | ✓ | 0.6804 | 0.6263 | 0.5645 |
| *Ours* | ✓ | ✓ | ✓ | **0.7656** | **0.6894** | **0.6051** |

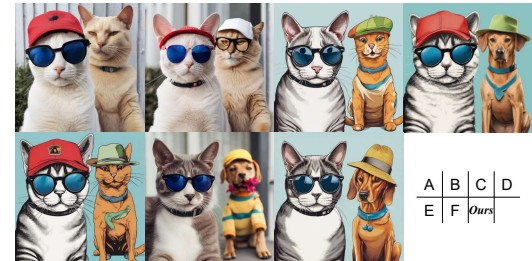

Figure 6: Text-to-Image generation with various configurations.

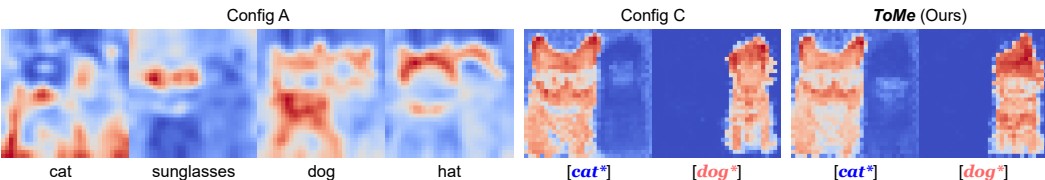

Config A     cat   sunglasses   dog   hat    Config C   [*cat**] [*dog**]    *ToMe* (Ours)   [*cat**] [*dog**]

Figure 7: Cross-Attention maps visualization with various configurations.

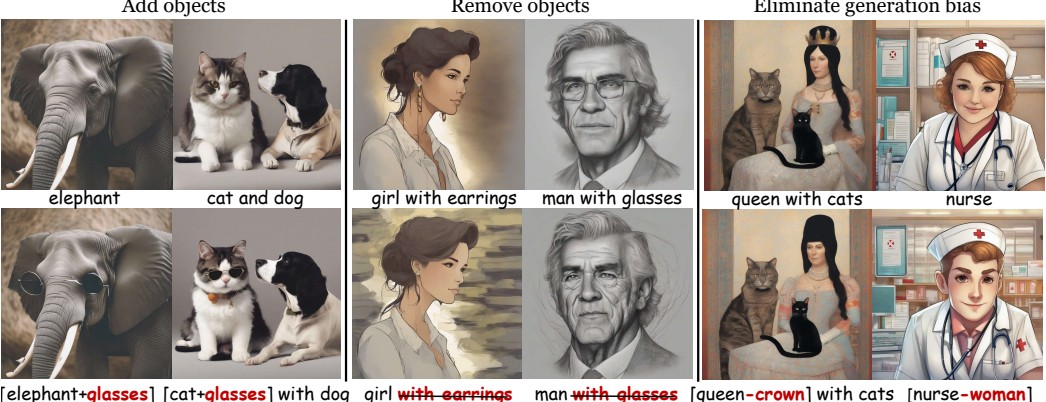

Add objects    Remove objects    Eliminate generation bias

elephant   cat and dog   girl with earrings   man with glasses   queen with cats   nurse

[elephant+**glasses**] [cat+**glasses**] with dog   girl ~~with earrings~~   man ~~with glasses~~   [queen-**crown**] with cats   [nurse-**woman**]

Figure 8: Additional applications of semantic additivity in text embedding.

approach *ToMe* shows improved performance in both object and attribute binding scenarios, which is consistent with the quantitative metrics reflected in Table 1.

**Ablation Study** over each component is quantitatively shown in Table 2. We can observe that using only token merging techniques (with *ToMe* and ETS as config.B) results in a slight performance improvement, which is consistent with the qualitative results in Fig. 6. However, token merging serve as the foundation for subsequent optimizations. When they are combined with the entropy loss $\mathcal{L}_{ent}$ as config.C, the performance improves significantly. We hypothesize that is partly due to the more regularized cross-attention maps as shown in Fig. 7. Nevertheless, conifg.C without the semantic binding loss still leads to worse generation performance in Fig. 6, as the dog on the right side still exhibits cat-like features. Incorporating the semantic alignment loss $\mathcal{L}_{sem}$ (as our default configuration) ensures that the two subjects correctly bind to their respective attributes without appearance confusion, achieving the best results quantitatively and qualitatively. Suppose token merging is ignored, and we only apply the optimization (Config D and Config E), the performances are only comparable to the baseline. Removing $\mathcal{L}_{ent}$ from *ToMe* (Config F) can also improve over the baseline, but the generation is with noticeable artifacts, which is mainly due to the less regularized cross-attention map. In conclusion, each element of these three novel techniques in *ToMe* contributes to achieving state-of-the-art performance. See Appendix D for more detailed ablation experiments.

**Additional Applications** of *ToMe* are shown in Fig. 8. *ToMe* can not only successfully address the semantic binding problem, it can also be applied to other problems widely exist in T2I generations, including adding objects [84, 70], removing objects [3, 22] and even bias mitigation [16, 61, 77, 78].

# 5 Conclusion

In this paper, we investigate a critical issue in text-to-image (T2I) generation models known as *semantic binding*. This phenomenon refers to instances where T2I models struggle to accurately interpret and visually bind the related semantics. Recognizing that previous methods often entail extensive fine-tuning of the entire T2I model or necessitate explicit specification of generation layouts by large language models, we introduce a novel training-free approach called Token Merging, denoted as *ToMe*, to tackle semantic binding issues in T2I generation. *ToMe* incorporates innovative techniques by stacking up the object token with its relevant tokens into a single *composite token*. This mechanism eliminate the semantic misalignment by unifying the cross-attention maps. Furthermore, we assist the *ToMe* with end token substitution, and iterative composite token updates technique to strengthen the semantic binding. In extensive experiments, we quantitatively compare it against various existing methods using the T2I-Compbench and our proposed GPT-4o benchmarks. The results demonstrate its ability to handle intricate and demanding generation tasks more effectively than current methods, especially for *object binding* cases that are ignored in previous research.

## Acknowledgements

We acknowledge project PID2022-143257NB-I00, financed by the Spanish Government MCIN/AEI/10.13039/501100011033 and FEDER. We acknowledge project "Science and Technology Yongjiang 2035" key technology breakthrough plan project (2024Z120). The Supercomputing Center of Nankai University supports computation.

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

# Appendix

## A   Limitations

Since our method is optimized for inference based on SDXL, it inherits some inherent limitations of SDXL. For example, it may produce artifacts in generated images and is unable to create images with complex layouts. Additionally, the *ToMe* technique relies on the CLIP text encoder to generate text embeddings, which may be subject to the limitations of the encoder itself. For instance, the CLIP encoder might not fully capture all the subtle semantic nuances in the text, which could restrict the performance of *ToMe* when processing certain types of text prompts. Addressing these limitations and advancing our understanding in these areas will help improve image generation technology.

## B   Broader Impacts

*ToMe* enhances the semantic binding capability in text-to-image synthesis by enhancing text embeddings. However, it also carries potential negative implications. It could be used to generate false or misleading images, thereby spreading misinformation. If *ToMe* is applied to generate images of public figures, it poses a risk of infringing on personal privacy. Additionally, the automatically generated images may also touch upon copyright and intellectual property issues.

## C   Implementation Details

### C.1   Method details

We extract the cross attention maps from the first three layers of the decoder in the UNet backbone, which contain rich semantic information, with a resolution of $32 \times 32$. For *Iterative composite Token Update*, since the early timesteps of the denoising process determine the layout of the image[27], we execute it only during the first 20% of the denoising process. All experiments were conducted on an NVIDIA-A40 GPU.

### C.2   Baseline methods implementation

For the quantitative comparison in Tab. 1, we used the official implementations of Ranni[21], ELLA[30], SyGen[58], and CoMat[34]. Since the SDXL versions of the Ranni[21], ELLA[30], and CoMat[34] methods have not been open-sourced, we refer to the BLIP-VQA scores reported in their respective papers. SynGen[58], like our method, performs optimization during inference. To ensure a fairer comparison, we adapted SynGen to SDXL.

### C.3   Text embedding analysis

Fig. 9's statistical analysis further demonstrates the information coupling property and semantic additivity of text embeddings. We employed MMDetection[12]and GLIP[38] to detect the probability of specified objects in images, referred to as *DetScore*, as shown in Fig. 9-(a). Fig. 9-(b) presents statistical results on 100 generated images, showing that the probability of detecting a hat in images generated from the text embedding corresponding to "a dog" is 0%. However, in images generated from the element-wise "[dog+hat]" additive embedding, the probability of detecting a hat is 68.61%, which is close to the probability of 73.12% for images generated using the prompt 'a dog wearing a hat'.

The information coupling of token embeddings is also reflected in the entropy of cross-attention for each token. Taking the prompt "a cat wearing sunglasses and a dog wearing a hat" as an example, we can extract the cross-attn map $\mathcal{A}_k \in \mathbb{R}^{1024}$ for each token, averaged over 50 time steps and multiple heads. After normalizing each map to 1.0(i.e., $\mathcal{A}_k[i] := \frac{\mathcal{A}_k[i]}{\sum_{i \in [1,32]} \mathcal{A}_k[i]}$), we calculate the token's infomation entropy as $\sum_{p_i \in A_k} -p_i \log(p_i)$. As shown in Fig. 9-(c), we conducted statistics on 100 generated images and found that tokens positioned later in the prompt tend to have higher entropy, indicating more dispersed cross-attn maps. This phenomenon might be attributed to CLIP's[56] masked attention mechanism, where each token can interact with all preceding tokens, and tokens

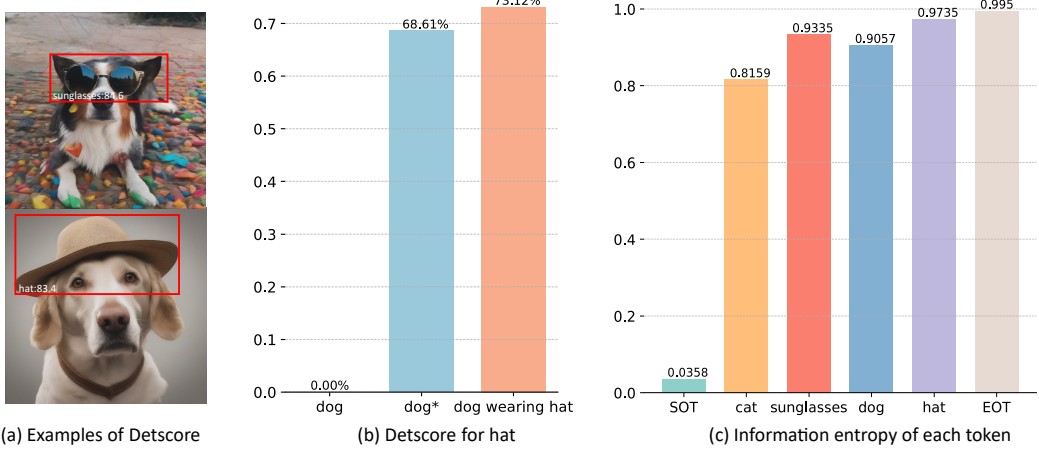

(a) Examples of Detscore     (b) Detscore for hat     (c) Information entropy of each token

Figure 9: Additional statistical analyses, all statistical values are averaged results from 100 images. (a) An example of DetScore visualization. (b) By fusing the dog and hat token, we obtain dog*, and the generated images often include a hat. The DetScore value for dog* is close to the DetScore value obtained using the complete prompt "a dog wearing a hat". (c) We calculated the entropy of the cross-attention maps for each token and found that tokens positioned later in the sequence generally have higher entropy, indicating that their cross-attention maps are more dispersed.

positioned later can interact with more tokens, thus containing more information. Consequently, we employ an entropy regularization loss to constrain each attention map to be as concentrated as possible, thereby reducing the amount of irrelevant information contained in each token embedding.

## C.4 Time complexity

Table 3: Time Complexity of various methods. The results of our method are highlighted in bold.

| Method | Inference Steps | Time Cost | Color | Texture | Shape |
|---|---|---|---|---|---|
| SDXL | 20 | 18s | 0.6136 | 0.5449 | 0.5260 |
| ToMe (Config C) | 20 | 23s | **0.7419** | **0.6581** | **0.5742** |
| ToMe (Ours) | 20 | 45s | **0.7612** | **0.6653** | **0.5974** |
| Ranni (SDXL) | 50 | 87s | 0.6893 | 0.6325 | 0.4934 |
| ELLA (SDXL) | 50 | 51s | 0.7260 | 0.6686 | 0.5634 |
| SynGen (SDXL) | 50 | 67s | 0.7010 | 0.6044 | 0.5069 |
| SDXL | 50 | 42s | 0.6369 | 0.5637 | 0.5408 |
| ToMe (Config C) | 50 | 56s | **0.7525** | **0.6775** | **0.5797** |
| ToMe (Ours) | 50 | 83s | **0.7656** | **0.6894** | **0.6051** |

Tab. 3 reports the inference time costs of various methods, all measured on a single NVIDIA-A40 GPU. We demonstrate that our method does not significantly increase inference time while improving semantic binding performance with 50 inference steps. We further extend this analysis by measuring the time cost with 20 inference steps and various ToMe configurations, as shown in the Tab. 3. We report the time cost (by seconds) along with BLIP-VQA scores across the color, texture, and shape attribute binding subsets. From this table, we can observe that using the token merging (ToMe) technique and entropy loss (Config.C), our method achieves excellent performance with minimal additional time cost. Additionally, even with only 20 inference steps, our method, ToMe, maintains high performance with very little degradation.

## C.5 GPT-4o Score

In order to better demonstrate the binding ability of our model for complex prompts. We have constructed a set of high-difficulty prompts, where the content primarily uses nouns to describe the subject. We use OpenAI's latest release, GPT-4o, to evaluate the quality of images generated by various models because GPT-4o excels in image discernment, allowing for precise evaluation of the generated outputs. As show in Fig. 10, We designed nine scoring levels, ranging from 0 to 100 points,

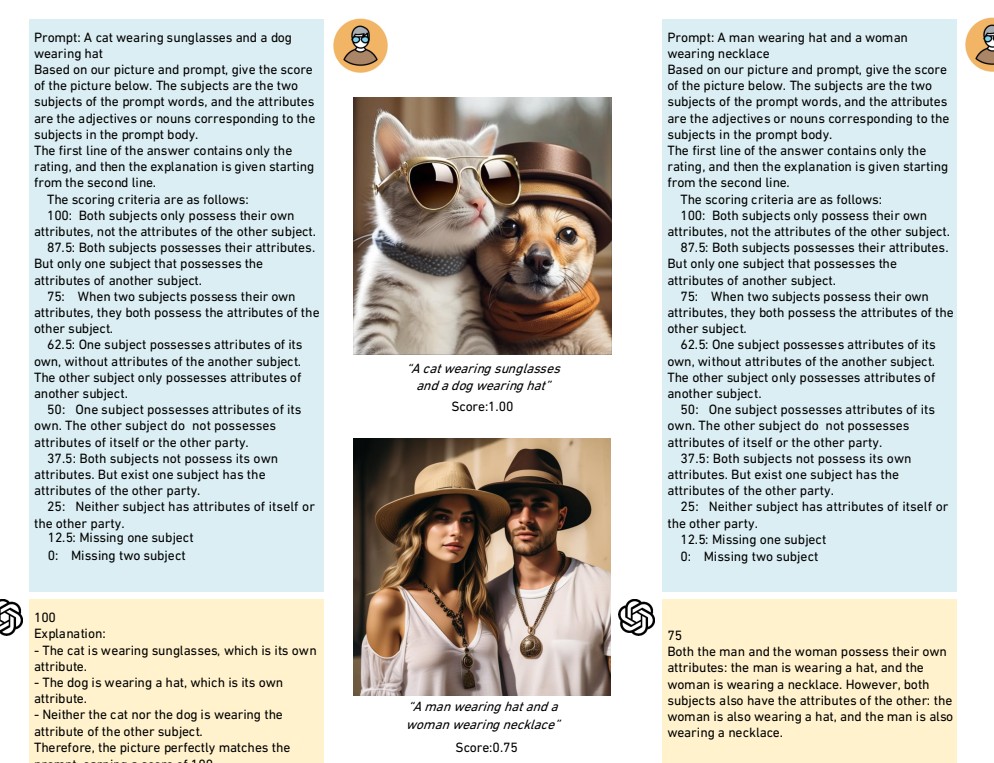

Figure 10: Evaluation Metric: GPT-4o

based on factors such as whether the objects correctly possess their attributes, the mixing of attributes between objects, and whether the objects are correctly generated, to distinguish different levels of generation quality.

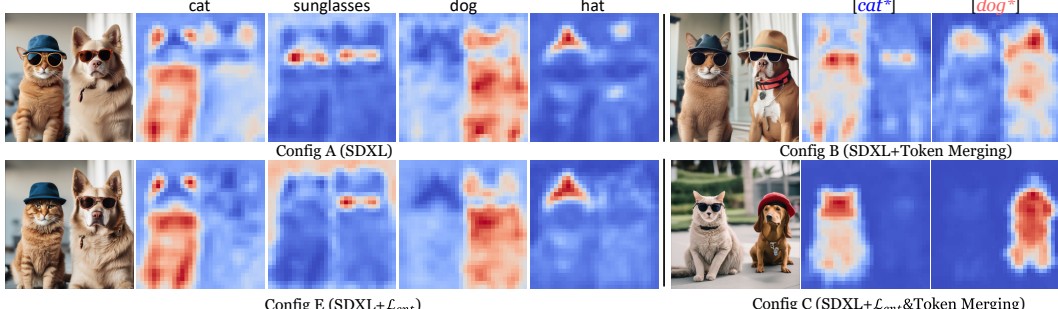

Figure 11: Cross-attention maps visualization with various configurations, with the input prompt "a cat wearing sunglasses and a dog wearing hat"

## D  Additional Ablation Studies

### D.1  More Configures and ETS ablation

As an example in Fig. 11, the original SDXL (Config.A) suffered from attribute binding errors due to divergent cross-attention maps. When only applying token merging (Config B), the co-expression of entities and attributes resulted in a dog wearing a hat in the image, but the attribute leakage issue remained due to the divergent cross-attention maps. When only applying the entropy loss $\mathcal{L}_{ent}$ (Config E), although the cross-attention maps corresponding to each token are more concentrated, they may focus on wrong regions. Only by applying both token merging and $\mathcal{L}_{ent}$ techniques (Config

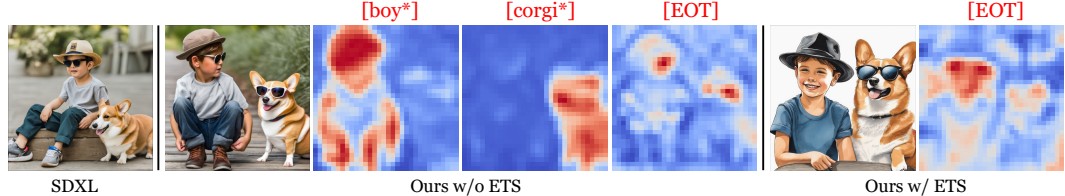

Figure 12: Ablation study of our proposed end token substitution (ETS) technique, with the input prompt "a boy wearing hat and a dog weairng sunglasses"

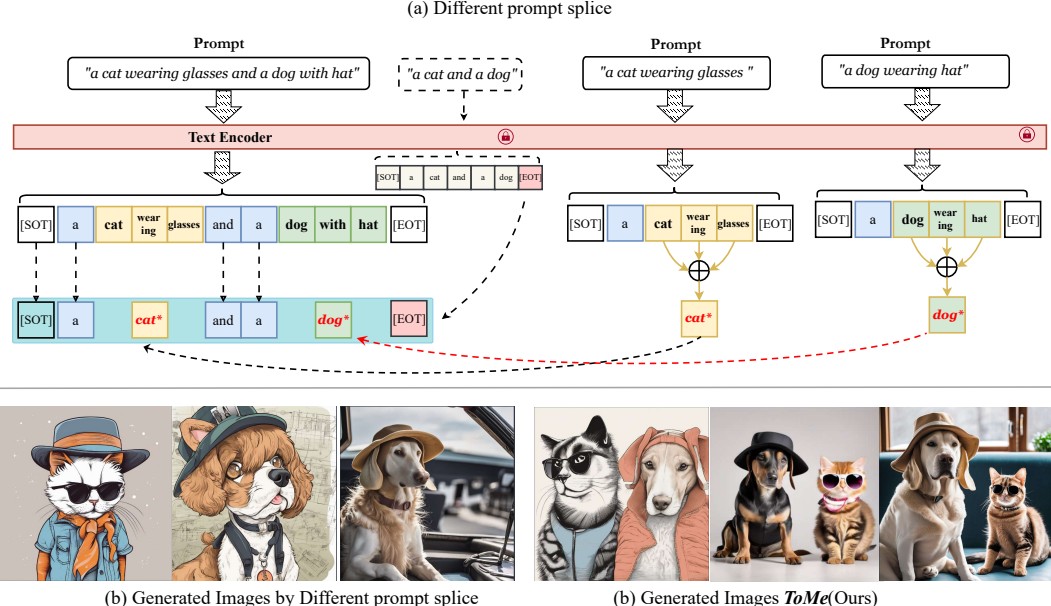

(b) Generated Images by Different prompt splice    (b) Generated Images *ToMe*(Ours)

Figure 13: Comparison of images generated by different prompts splice

C), the cross-attention map of the composite token becomes better concentrated on the correct areas and thus leading to more satisfactory semantic binding of entities and attributes.

The end token substitution (ETS) technique is proposed to address potential semantic misalignment in the final tokens of long sequences. As the [EOT] token interacts with all tokens, it often encapsulates the entire semantic information, as shown in Fig. 2. Therefore, the semantic information in [EOT] can interfere with attribute expressions, we mitigate this by replacing [EOT] to remove the attribute information it contains from the original prompts, retaining only the semantic information for each subject.

For example, as the cross-attention maps and T2I generation performance shown in Fig.12, when ToMe is not combined with the EST technique, the 'sunglasses' semantics contained in the EOT token cause the boy to incorrectly wear sunglasses. However, when combined with ETS, the unwanted semantic binding is relieved.

## D.2    Different prompts splice

In Sec. 3.2.1, we fuse each object and its corresponding attributes. At this stage, both the object token embedding and the attribute token embedding are derived from the text embedding obtained by processing the same prompt through the CLIP Text Encoder, potentially causing the information between them to be coupled. We also experimented with splicing token embeddings from different prompts, as illustrated in Fig. 13. While keeping other components of *ToMe* unchanged, the resulting images often exhibit a missing of the object. We hypothesize that this may be due to the lack of contextual semantics between token embeddings from different prompts[8].

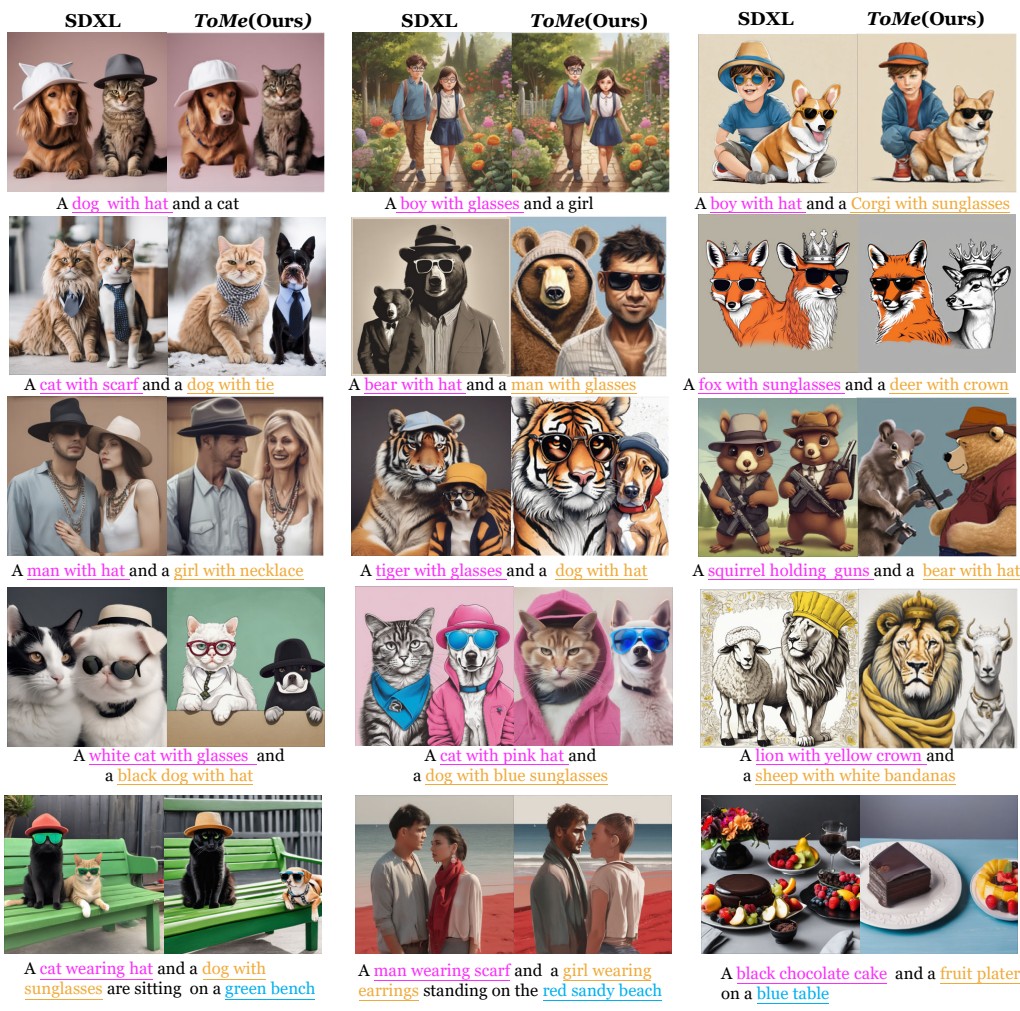

Figure 14: Additional semantic binding results. Our method not only achieves good results in object binding but is also effective for composite binding of objects and their adjective attributes.

# E  Additional Results

As shown in Tab. 4, we have added quantitative comparison results with additional methods. Our method consistently outperforms or is on par with the existing methods. Fig. 14 presents more qualitative comparison results, demonstrating that our method achieves good performance in attribute binding, object binding, and the composite binding of attribute and objects. *ToMe* can also generate images with subjects or backgrounds featuring multiple attributes(Fig. 14, the last line), in this scenario, we find that using an additional positional loss[19] based on the attention map is effective.

We also conduct a user study with 20 participants to enrich the evaluation. Here we compare our method ToMe with SDXL[53], SynGen[58], Ranni[21] and ELLA[30]. As shown in Fig. 15, we ask the participants to rate the semantic binding into 4 levels and calculate the distribution of each comparison method over these four diverse levels. We can observe that our method better achieve the semantic binding performance by mainly distribute in the highest level 1, while the other methods struggle to obtain user satisfactory results.

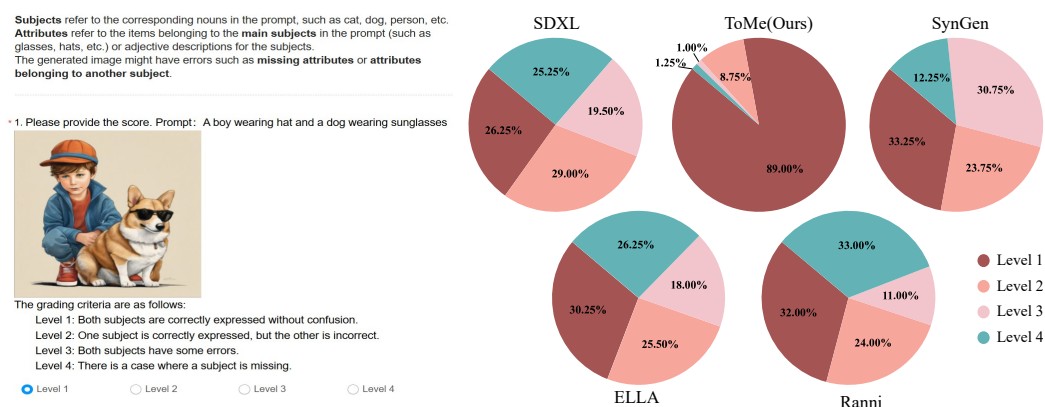

Figure 15: User study with 20 participants, we ask users to rate the semantic binding into four levels.

Table 4: Comparison of BLIP-VQA Scores

| Method | Base Model | Train | BLIP-VQA ↑ | | |
|---|---|---|---|---|---|
| | | | Color | Texture | Shape |
| SD v1.5[60] | - | ✓ | 0.3750 | 0.4159 | 0.3724 |
| SD v2[60] | - | ✓ | 0.5065 | 0.4922 | 0.4221 |
| DALL-E2[57] | - | ✓ | 0.5750 | 0.6374 | 0.5464 |
| SDXL[53] | - | ✓ | 0.6369 | 0.5637 | 0.5408 |
| PlayG-v2[37] | - | ✓ | 0.6208 | 0.6125 | 0.5087 |
| Ranni[21] | SD1.5 | ✓ | 0.2414 | 0.3029 | 0.2857 |
| ELLA[30] | SD1.5 | ✓ | 0.6911 | 0.6308 | 0.4938 |
| SynGen[58] | SD1.5 | ✗ | 0.6619 | 0.6451 | 0.4666 |
| CoMat[34] | SD1.5 | ✓ | 0.6561 | 0.6190 | 0.4975 |
| Composable v2[45] | SD2.0 | ✗ | 0.4063 | 0.3645 | 0.3299 |
| Structured v2[20] | SD2.0 | ✗ | 0.4990 | 0.4900 | 0.4218 |
| Attn-Exct v2[7] | SD2.0 | ✗ | 0.6400 | 0.5963 | 0.4517 |
| GORS[31] | SD2.0 | ✗ | 0.6603 | 0.6287 | 0.4785 |
| Ranni[21] | SDXL | ✓ | 0.6893 | 0.6325 | 0.4934 |
| ELLA[30] | SDXL | ✗ | 0.7260 | 0.6686 | 0.5634 |
| SynGen[58] | SDXL | ✗ | 0.7010 | 0.6044 | 0.5069 |
| CoMat[34] | SDXL | ✓ | 0.7774 | 0.6591 | 0.5262 |
| *ToMe* (Ours) | SDXL | ✗ | 0.7656 | 0.6894 | 0.6051 |

