# OpenReview forum: "Token Merging for Training-Free Semantic Binding in Text-to-Image Synthesis"
_NeurIPS.cc/2024/Conference — NeurIPS 2024 poster_

### Official Review · Reviewer_8gpY · 2024-06-19

**Soundness:** 2
**Presentation:** 2
**Contribution:** 3
**Rating:** 7
**Confidence:** 4

**Summary:**

The authors tackle semantic binding, where T2I models often fail to correctly reflect the relations between objects (object binding) or objects and their attributes (attribute binding). To address this, they introduce Token Merging (ToMe), a method that aggregates related tokens to a single composite token, which ensures they share the same cross-attention map. To do that, two training-free optimizations are introduced. The first, Semantic Binding loss, which makes sure the composite token leads to noise prediction that is consistent with the full phrase the token is based on. The second, Entropy loss, which helps the tokens focus exclusively on their designated regions.

**Strengths:**

* Creating a composite token and then optimizing it is a creative and elegant approach to the problem. I'm curious to see what future work can be built on top of this contribution.
* The focus on object binding is indeed missing from the literature, a topic that is well-addressed in this work.
* The idea of end token substitution to remove attribute information is simple.

**Weaknesses:**

* The related work underplays the role of Attend-and-Excite and Linguistic Binding in Diffusion Models (SynGen) have on this paper. The former introduced the idea of semantic guidance, while the latter used dependency graphs to extract syntactically related tokens from prompts and presented a loss that encourages cross-attention maps of syntactically related tokens to agree. Both of which are ideas this paper meaningfully builds on.

* You do not describe how you obtain the syntactic information: entities and their relevant objects nor if it is automatic or an input constraint. This is particularly confusing because in section 3.2 you begin describing exactly that, but do not delve into the specifics. Furthermore, in section 4.1 (line 248), you mention spaCy and that you "identify each object and its corresponding attributes for token merging", but do not provide actual details about your method. Does it mean it can accurately capture *all* syntactic structures? It is possible that I misunderstand, but it feels like the syntactic analysis portion of this work borrows from the SynGen paper. If this is the case, give appropriate credit. If it is not, then add details, as this is part of the method and is quite unclear.

* If I understand correctly, there are no human evaluation experiments, a user study would provide merit to ToMe’s superior performance. As a side note, in table 1, ‘Human-preference’ is a slightly misleading title. If I understand correctly, it is the output of the Image Reward human-preference model, but it sounds as if it is a human-evaluation result.

* I find that despite the claim that ToMe works well on highly complex prompts, there are no such examples or experiments given. For example, Figure 5 depicts prompts that are supposed to be complex, but they are no different from the simplistic prompts in Attend-and-Excite (“a {color_1} {subject_1} and {a color_2} {subject_2}”), only that they address object binding.

**Questions:**

1. What input text is used in the global encoding part?
2. Do you expect your method to extend to SD-3?
3. I worry that ToMe suffers from too many gradient updates in the event of highly complex prompts (like all training-free approaches), which would push the latent out of distribution.
4. Does ToMe work on long captions, with multiple sentences? How?
5. You say in the appendix that SynGen was modified to work with SDXL. Can you elaborate on the modifications? What would be needed to reproduce this experiment too?

**Limitations:**

See questions 3 and 4.

---

> ### Author Rebuttal · Authors · 2024-08-06
>
> We appreciate your feedback and will incorporate the discussions mentioned below to enhance the quality of our paper. Note that we utilize the numerical references to cite sources within the main paper.
>
> **W1: Related work**
>
> These two methods generally fall under optimization-based approaches. They primarily adjust noisy signals to enhance attention maps and strengthen semantic binding. Attend-and-Excite improves object presence by increasing the attention score for each object. SynGen performs a syntactic analysis of the prompt to identify entities and their modifiers, using attention loss functions to ensure that cross-attention maps align with the linguistic binding reflected in the syntax. These methods have indeed provided a solid foundation for future semantic binding work. Different from these two works, our training-free method ToMe is based on the text embedding additivity property. Instead of updating the latent space, we update the composite token embeddings. We further introduce two auxiliary losses, i.e. the entropy loss and semantic binding loss, to augment the ToMe performance.
> We will include more detailed introductions to these two methods in the introduction and related work sections in any future version.
>
> **W2: Syntactic information**
>
> In Section 4.1 on the implementation details, we mention that we use SpaCy[27] to detect relationships, following the approach outlined in the SynGen. More specifically, we use SpaCy’s transformer-based dependency parser to analyze the prompt and identify all entity nouns, including both proper nouns and common nouns, that are not serving as direct modifiers of other nouns. This process helps pinpoint entity nouns and their corresponding modifiers. We will further detail this process in the method and implementation details section.
>
> **W3: User study**
>
> As discussed in the General Response 1. In this paper, we use the ImageReward[72] model to evaluate human preference scores, which measures image quality and prompt alignment. However, since quantitative results may not fully capture these aspects, we now conduct a user study with 20 participants to enrich the evaluation. Here we compare our method ToMe with SDXL, SynGen, Ranni and ELLA. Each participant is questioned with 100 images, each generated from five methods and twenty prompts.  The results are presented in Fig.16 of the rebuttal file. In the user study, we ask the participants to rate the semantic binding into 4 levels. We then calculate the distribution of each method over the four diverse levels. We can observe that ToMe better achieve the semantic binding performance by mainly distribute in the highest level 1, while the other methods struggle to obtain satisfactory results.
>
> **W4&Q4: Complex prompts generation**
>
> As illustrated in General Response 2, ToMe is not limited to only simple cases. In the main paper Table 2, we show experiments on the T2I-CompBench benchmark, where there are long captions with various prompt templates. There are also complex prompts T2I generations in Fig.12. To further show that our method ToMe can be applied to the T2I generation with complex prompts. We adopt the suggestions from the reviewer to generate with more complicated prompt templates, that includes longer prompts with multiple objects, multiple attributes or even multiple sentences. The generation results are shown in Fig.13 in the rebuttal file.
>
>
> **Q1: Global encoding text**
>
> SDXL is using a conditioning concatenation for text embeddings with two prompts. To keep our method ToMe applicable and generalizable to other T2I generative models, we keep both prompts the same. For example, with prompt “A blue cat and a green dog”, we assign this prompt as both the local and global encoding prompt and input it to the two text encoders and concatenate the text embeddings along the channels. And in our method, ToMe is updating the concatenated text embedding conditions with our proposed losses. We will add this to the implementation details.
>
> **Q2: Generalizability to other T2I models**
>
> As claimed in General Response 4, T2I models are facing generation limitations for various specific cases. We aim, as a future goal, to generalize our method to various generative models (such as DeepFloyd-IF, SD3, PixArt, etc.) to counter this issue. Current T2I generation models are built on various language models and vision-language models, such as CLIP and T5, and incorporate diverse architectures, including UNet, GAN, and DiT. We are also curious to see if the additivity property truly exists across all these cases, and whether this property is a generalizable feature.
>
>
> **Q3: Latent Distribution Drifts**
>
> You are correct; large gradients can cause significant changes in the latent space, which is a common issue in training-free semantic binding methods[7,41,56]. The primary goal of these methods is to update the latent or text embeddings to align the semantics between texts and images, often resulting in varying degrees of distribution change. To mitigate this issue, we only update the composite token representations during the first $T_{\text{opt}} = 0.2T$ time steps. This technique helps our method ToMe experience fewer distribution changes from the base model SDXL, as demonstrated in multiple cases in Fig. 5, Fig. 12 and Fig.13.
>
> **Q5: SynGen reproduction details**
>
> The original SynGen paper is based on the SD1.5 model. To adapt it to use SDXL as the base model, we utilize the 32×32 cross-attention map from the first three layers of the UNet-decoder, which serves as the most semantic layer, similar to the 16×16 cross-attention in SD1.5. We also performed a grid search to determine the optimal learning rate for SynGen in this context. All other details remain consistent with the original paper. After tuning the hyperparameters, the SDXL-based SynGen outperforms the SD1.5-based SynGen, as shown in Table 1 and Table 4. We will release the SynGen adapted code in the future for reproduction.

---

> > ### Author Response · Authors · 2024-08-13
> > **Thanks for your comments and look forward to further discussion**
> >
> > Dear Reviewer 8gpY:
> >
> > Thank you for your valuable feedback on our paper. We deeply appreciate the time and effort you’ve put into reviewing our work, and your constructive comments are invaluable to us.
> >
> > Regarding the weaknesses you mentioned, we include the corresponding experiments and discussion as follows:
> >
> > 1. Expanded discussion on related work, particularly Attend-and-Excite and SynGen.
> >
> > 2. Added detailed explanations on syntactic information extraction using SpaCy.
> >
> > 3. Conducted a user study with 20 participants to validate ToMe’s performance.
> >
> > 4. Provided additional examples of complex prompts generation.
> >
> > 5. Clarified the input text used in global encoding.
> >
> > 6. Discussed the generalizability of our method to other T2I models.
> >
> > 7. Discussed the potential latent distribution drifts.
> >
> > 8. Detailed the modifications made to adapt SynGen for SDXL.
> >
> > Your feedback is incredibly important to us, and we sincerely thank you for considering our rebuttal. Please let us know if there are any further concerns or questions—we are more than happy to discuss them.
> >
> > Thank you again for your time, and we look forward to your response.
> >
> > Best regards,
> > Authors of submission 1763

---

> > ### Comment · Reviewer_8gpY · 2024-08-13
> > **Thank you**
> >
> > Thank you for taking the time to write back and answering some of my questions.
> >
> > **W2** Note that the current 4.1 section does not mention the SynGen paper, but I trust that this will be clarified in a revised version.
> >
> > **Q1** Please add it to the paper.
> >
> > **Q2** Great. For your consideration: it would be interesting to see a FLUX or SD-3 implementation in a revised manuscript.
> >
> > **Q3** Thanks, please add that to the paper, as these optimization methods are indeed quite sensitive, and understanding the tradeoffs is important.
> >
> > **Q5** Please specify the hyperparameters that were used and any other detail that can be used to reproduce your calibration of this baseline.
> >
> > There are lots of details that were not provided in the discussion, but I trust that you will incorporate everything in great detail in a revised manuscript. I'm raising my score to reflect this.

---

> > > ### Author Response · Authors · 2024-08-13
> > >
> > > Thank you very much for reviewing our paper and providing valuable feedback. We're glad our rebuttal addressed your concerns, and we'll include the suggested experiments and detailed discussion in the revised manuscript. Thanks again for your time and effort.

---

### Official Review · Reviewer_85Q2 · 2024-07-10

**Soundness:** 3
**Presentation:** 4
**Contribution:** 3
**Rating:** 6
**Confidence:** 4

**Summary:**

The authors propose a method to mitigate semantic binding, a common phenomenon in text-to-image models. While previous methods explicitly control the attention maps so that nouns and attributes attend to the same regions, the authors propose combining the nouns and attributes into a single token. This approach enforces the attention maps to concentrate in the same area. Additionally, they introduce an inference-time optimization loss to ensure the attention maps are focused, and that each composed token retains the same semantic meaning as all the different tokens that comprise it. The authors also analyze the information encoded in different tokens at the output of the text encoder.

**Strengths:**

1. The authors present an interesting analysis of the information encoded tokens, specifically they show that much of the semantic information is also encoded in the EOT tokens.
2. The authors provide a method to mitigate semantic binding in prompts of the form: “A <noun_A> <Attribute_A> and <noun_B> <Attribute_B>”, that outperforms or performs comparably to existing methods across various benchmarking subsets.

**Weaknesses:**

1. It is not clear how the model performs on more complicated prompts where both the noun and its sub-objects have attributes, such as: “A blue cat wearing sunglasses and a yellow dog wearing a pink hat.”
2. The method is on the slower side compared to other existing methods, due to inference time optimization.
3. This work is missing a user study, as the proposed automatic metrics can be inaccurate, and human evaluation is common in previous works.

**Questions:**

1. How does the method perform on more complex prompts? For example, for a prompt like “A blue cat wearing sunglasses and a yellow dog wearing a pink hat”. From my understanding, the method should not succeed in separating “pink hat” and “yellow dog” as the whole expression will be combined into a single token.
2. Is it not clear to me what prevents semantic binding across nouns, e.g. both the dog* and the cat* tokens to have the same appearance, (e.g. a cat-dog hybrid), which is a common problem in text-to-image models. What prevents the dog* and cat* attention maps from attending to the same regions?

**Limitations:**

The authors propose limitations relating to the underlying models, but I would be interested to learn in which areas of semantic binding the paper still struggles, such as with more complex prompts or a larger number of nouns, etc..

---

> ### Author Rebuttal · Authors · 2024-08-06
>
> We appreciate your feedback and will incorporate the discussions mentioned below to enhance the quality of our paper. Note that we utilize the numerical references to cite sources within the main paper.
>
> **W1&Q1&L1: Complex prompts generation**
>
> As demonstrate in General Response 2, our method ToMe is not limited to only simple cases. In the main paper Table 2, we show experiments on the T2I-CompBench benchmark, where there are long captions with various prompt templates. There are also complex prompts T2I generations in Fig.12 in the Appendix.  To further show that our method ToMe can be applied to the T2I generation with complex prompts. We adopt the complicated long caption templates suggested from the reviewer. The generation results are shown in Fig.13 in the rebuttal file.
>
> In this case, we use our method ToMe in an iterative updating way. As an instance, we generate the fourth image example in Fig.13 (left down) with the prompt “a blue cat wearing sunglasses and a yellow dog wearing a pink hat”. To apply ToMe, we merge ‘yellow dog’ into token1 (dog*) and ‘pink hat’ into token2 (hat*). Following that, we merge token1 and token2 into token3. The noun phrases for computing the semantic binding loss corresponding to these three tokens are ‘a yellow dog’, ‘a pink hat’ and ‘a dog wearing a hat’, respectively. Then we update token1 and token2 by our losses proposed in Section 3.2.2.
>
> **W2: Inference time cost**
>
> As detailed in the General Response 3, we present a time cost comparison in Table 3 in Appendix C.4, where we compare performance using 50 inference steps using the SDXL model with float32 version on an A40 GPU.  We demonstrate that our method does not significantly increase inference time while improving semantic binding performance. We further extend this analysis by measuring the time cost with 20 inference steps and various ToMe configurations, as shown in the Table below. We report the time cost (by seconds) along with BLIP-VQA scores across the color, texture, and shape attribute binding subsets. From this table, we can observe that using the token merging (ToMe) technique and entropy loss (Config.C in Table 2), our method achieves excellent performance with minimal additional time cost. Additionally, even with only 20 inference steps, our method, ToMe, maintains high performance with very little degradation.
>
> | method | inference steps | Time Cost | Color | Texture | Shape |
> | :--------: | :--------: |:--------: |:--------: |:--------: |:--------: |
> SDXL                    |  20        |  18s  |  0.6136 |  0.5449  |  0.5260 |
> *ToMe (Config C)*  |   20    |   23s   |   *0.7419*  |  *0.6581*  |   *0.5742* |
> **ToMe (Ours)**         |   20    |   45s   |   **0.7612**  |   **0.6653**   |  **0.5974**  |
> Ranni (SDXL)     |  50    |  87s    |  0.6893   |  0.6325   |  0.4934 |
> ELLA (SDXL)     |  50    |  51s    |  0.7260   |  0.6686   |  0.5634 |
> SynGen (SDXL)   |  50    |  67s    |  0.7010   |  0.6044   |  0.5069 |
> SDXL                    |  50    |  42s    |  0.6369   |  0.5637   |  0.5408 |
> *ToMe (Config C)* |   50    |   56s   |   *0.7525*  |  *0.6775*  |   *0.5797* |
> **ToMe (Ours)**         |   50    |   83s   |   **0.7656**  |   **0.6894**   |   **0.6051**  |
>
>
> **W3: User study**
>
> Please also refer to the General Response 1. In this paper, we use the ImageReward[72] model to evaluate human preference scores, which comprehensively measures image quality and prompt alignment. However, since quantitative results may not fully capture these aspects, we now also conduct a user study with 20 participants to enrich the evaluation. Here we compare our method ToMe with SDXL[51], SynGen[56], Ranni[19] and ELLA[28]. Each participant is questioned with 100 images, each generated from these five methods and twenty prompts.
>
> The results are presented in Fig.16 of the rebuttal file. Similar to our proposed GPT-4o benchmark (detailed in Appendix C.5 and Fig.10), we ask the participants to rate the semantic binding into 4 levels and calculate the distribution of each comparison method over these four diverse levels. We can observe that our method better achieve the semantic binding performance by mainly distribute in the highest level 1, while the other methods struggle to obtain user satisfactory results.
>
>
> **Q2: Mechanism to avoid semantic misalignment**
>
> In this paper, we propose two losses to avoid the semantic misalignment and cross-attention leakage among objects and attributes.
>
> First, the semantic binding loss assist in purifying the token semantics. Building on the ToMe technique, we decrease the semantic binding loss to enhance the cross-attention. It encourages the newly learned token to infer the same noise prediction as the original corresponding phrase, reinforcing the semantic coherence between the text and the generated image. To ensure that the semantics of the composite tokens accurately correspond to the noun phrases they represent, we use a clean prompt as a supervisory signal. The semantic binding loss is computed separately, ensuring no interference among noun words.
>
> In addition, we reduce the entropy loss to help ensure that tokens focus exclusively on their designated regions, preventing the cross-attention maps from becoming overly divergent. The experimental results comparing scenarios with and without entropy loss are presented in Table 2, Fig. 6, and Fig. 7. Both losses are functioning together to improve the semantic binding.

---

### Official Review · Reviewer_mo93 · 2024-07-11

**Soundness:** 3
**Presentation:** 3
**Contribution:** 3
**Rating:** 6
**Confidence:** 3

**Summary:**

This paper aims to solve the semantic binding problem in T2I models. The authors introduce a semantic binding method by merging tokens of entities and related attributes. Besides, several other tricks like semantic binding loss and entropy loss are introduced to improve the performance of semantic binding.

**Strengths:**

- The idea is straightforward and it is also reasonable that it will work with some other tricks.
- The paper is well-written.

**Weaknesses:**

- The importance of a few tricks like entropy loss is not well-emphasized.
- The motivation for some tricks like substituting the EOT token is not well-explained.

**Questions:**

- 1. From Table 2, we know that entropy loss + token merging bring the biggest impact. This is reasonable because token merging actually changes nothing, all information still exists in the composition token, and semantic leakage will also happen. However, the reviewer does not fully understand what happens when doing token merging + entropy loss. Why do they work so well? Actually, we don't change the information and the way the information interacts.
    - 1.1. The reviewer thinks Figure 7 is quite clear for the motivation, considering entities and their related attributes as one object. The reviewer is confused about what function entropy loss works.

- 2. What is the impact of substituting the EOT token? Is it important in the whole process?

- 3. How about the efficiency compared to the baseline in Table 1?

**Limitations:**

Yes

---

> ### Author Rebuttal · Authors · 2024-08-06
>
> We appreciate your feedback and will incorporate the discussions mentioned below to enhance the quality of our paper. Note that we utilize the numerical references to cite sources within the main paper.
>
> **W1&Q1: Entropy loss**
>
>  In Appendix C.3, we demonstrate that the information coupling of token embeddings is also reflected in the entropy of cross-attention for each token. In Fig. 9-(c), we calculated the entropy of cross-attention maps for each token and found that tokens appearing later in the sequence generally have higher entropy, indicating that their cross-attention maps are more dispersed.
>
> Based on this observations, in this paper, we combine the entropy loss with our proposed ToMe approach. Reducing the entropy of these cross-attention maps of composite tokens helps to ensure that these tokens focus exclusively on their designated regions, preventing the cross-attention maps from becoming overly divergent, thus avoiding cross-attention leakage [64]. The experimental results comparing scenarios with and without entropy loss are presented in Table 2, Fig.6, Fig.7 and Fig.14 (in the rebuttal file).
>
> As an example in Fig.14, the original SDXL (Config A) suffered from attribute binding errors due to divergent cross-attention maps. When only applying token merging (Config B), the co-expression of entities and attributes resulted in a dog wearing a hat in the image, but the attribute leakage issue remained due to the divergent cross-attention maps. When only applying the entropy loss (Config E), although the cross-attention maps corresponding to each token are more concentrated, they may focus on wrong regions. Only by applying both token merging and $\mathcal{L}_{ent}$ techniques (Config C), the cross-attention map of the composite token becomes better concentrated on the correct areas and thus leading to more satisfactory semantic binding of entities and attributes.
>
> **W2&Q2: EOT ablation study**
>
> The end token substitution (ETS) technique is proposed to address potential semantic misalignment in the final tokens of long sequences. As the [EOT] token interacts with all tokens, it often encapsulates the entire semantic information, as shown in Fig. 2. Therefore, the semantic information in [EOT] can interfere with attribute expressions, we mitigate this by replacing [EOT] to remove the attribute information it contains from the original prompts, retaining only the semantic information for each subject.
>
> For example, as the cross-attention maps and T2I generation performance shown in Fig.15 (in the rebuttal file), when ToMe is not combined with the EST technique, the ‘sunglasses’ semantics contained in the EOT token cause the boy to incorrectly wear sunglasses. However, when combined with ETS, the unwanted semantic binding is relieved.
>
> **Q3: Efficiency**
>
> As detailed in the General Response 3, we present a time cost comparison in Table 3 in Appendix C.4, where we compare performance using 50 inference steps using the SDXL model with float32 version on an A40 GPU. We demonstrate that our method does not significantly increase inference time while improving semantic binding performance. We further extend this analysis by measuring the time cost with 20 inference steps and various ToMe configurations, as shown in the Table below. We report the time cost (by seconds) along with BLIP-VQA scores across the color, texture, and shape attribute binding subsets. From this table, we can observe that using the token merging (ToMe) technique and entropy loss (Config.C in Table 2), our method achieves excellent performance with minimal additional time cost. Additionally, even with only 20 inference steps, our method, ToMe, maintains high performance with very little degradation.
>
> | method | inference steps | Time Cost | Color | Texture | Shape |
> | :--------: | :--------: |:--------: |:--------: |:--------: |:--------: |
> SDXL                    |  20        |  18s  |  0.6136 |  0.5449  |  0.5260 |
> *ToMe (Config C)*  |   20    |   23s   |   *0.7419*  |  *0.6581*  |   *0.5742* |
> **ToMe (Ours)**         |   20    |   45s   |   **0.7612**  |   **0.6653**   |  **0.5974**  |
> Ranni (SDXL)     |  50    |  87s    |  0.6893   |  0.6325   |  0.4934 |
> ELLA (SDXL)     |  50    |  51s    |  0.7260   |  0.6686   |  0.5634 |
> SynGen (SDXL)   |  50    |  67s    |  0.7010   |  0.6044   |  0.5069 |
> SDXL                    |  50    |  42s    |  0.6369   |  0.5637   |  0.5408 |
> *ToMe (Config C)* |   50    |   56s   |   *0.7525*  |  *0.6775*  |   *0.5797* |
> **ToMe (Ours)**         |   50    |   83s   |   **0.7656**  |   **0.6894**   |   **0.6051**  |

---

> ### Author Response · Authors · 2024-08-13
> **Thanks for your comments and look forward to further discussion**
>
> Dear Reviewer mo93:
>
> Thank you for your valuable feedback on our work. Your constructive comments on our work are invaluable, and we genuinely hope to get feedback from you.
>
> Regarding the weaknesses you mentioned, we include the corresponding experiments and discussion as follows:
>
> 1. Entropy Loss & Token Merging: Our experiments (Fig. 14 in the rebuttal file) show that combining these techniques effectively focuses cross-attention maps, improving semantic binding.
>
> 2. End Token Substitution (ETS): As illustrated in Fig. 15 in the rebuttal file, ETS is also can prevent unintended semantic interference, leading to more accurate attribute expressions.
>
> 3. Efficiency Compare: We've provided a time cost analysis , showing that our method improves performance with minimal additional time.
>
> Your feedback is incredibly important to us, and we sincerely thank you for considering our rebuttal. We are more than happy to discuss them if you have any further concerns or questions.
>
> Thank you again for your time and effort to review our work and looking forward to your response.
>
> Best Regards,
>  Authors of submission 1763

---

> > ### Comment · Reviewer_mo93 · 2024-08-13
> > **Discussion**
> >
> > Dear authors:
> >
> > Thanks for your response. They address my concerns.
> >
> > I will raise my score.
> >
> > Best,
> >
> > Reviewer mo93

---

> > > ### Author Response · Authors · 2024-08-13
> > >
> > > Thank you very much for taking the time to review our paper and offering insightful feedback. We're pleased that our response addressed your concerns, and we will incorporate the recommended experiments and thorough discussion in the revised manuscript. We truly appreciate your time and effort.

---

### Official Review · Reviewer_jwvK · 2024-07-21

**Soundness:** 3
**Presentation:** 3
**Contribution:** 3
**Rating:** 6
**Confidence:** 3

**Summary:**

This paper focuses on the problem of lack of semantic binding in text-to-image generation models, and specifically on the misalignment between objects and their sub-objects. The paper introduces a training-free T2I method named ToMe after analyzing the properties of CLIP text embeddings and diffusion models. Utilizing the composition recognition ability of diffusion models, ToMe binds the embeddings of objects with their associated sub-objects together, and retains only the semantic information in the [EOT] tokens. ToMe is shown to be simple yet effective from the experimental results.

**Strengths:**

1. The paper is well-written and easy to follow.
2. The authors present detailed analysis of the CLIP text embeddings including their coupled and entangled properties and the semantically additive properties of text embeddings shown in diffusion models.
3. The structure of the proposed ToMe model is training-free, simple and mostly clear.
4. The authors provide extensive experimental results including quantitative, qualitative and ablation studies on the T2I-CompBench and GPT-4o Benchmark (introduced in this paper).

**Weaknesses:**

1. The Iterative composite Token Update (Sec. 3.2.2) is not clear enough to me. The authors introduce two losses: semantic binding loss and entropy loss. And since ToMe is training-free, it is not clear in the paper how these losses help the update the tokens at test time.

**Questions:**

1. Can the proposed method generalize to more objects, or background with various properties? It would be great if the authors can explain how generating more objects will change the pipeline, or if this is not feasible, please give an explanation why.
2. Please demonstrate how the two losses are integrated in the T2I pipeline, and how they are used to update the tokens.

**Limitations:**

The authors presented the inherent limitation of SDXL that ToMe is based on like producing artifacts in generated images, and unable to generate images with complex layouts. Also the limitation of CLIP to produce text embeddings could restrict the performance of ToMe.

---

> ### Author Rebuttal · Authors · 2024-08-06
>
> We appreciate your feedback and will incorporate the discussions mentioned below to enhance the quality of our paper. Note that we utilize the numerical references to cite sources within the main paper.
>
> **W1&Q2: Token Update**
>
> By training-free, we mean that our method ToMe does not involve training over datasets, unlike finetuning-based methods such as Ranni[19], ELLA[28], and CoMat[32]. Like other works (AnE[7], D&B[41], SynGen[56], etc.), methods that only backpropagate during inference time (on the generation of a single image) are considered to be training-free.
>
> ToMe distinguishes itself from other optimization-based methods by updating token embeddings, whereas most existing methods (e.g., AnE[7], D&B[41], SynGen[56]) update the latent space using designed loss functions. Instead of back-propagating the gradients to the T2I model parameters or the latent variables of the previous timestep latent $z_t$, we compute the loss and apply the gradients to the composite token embeddings in each inference step during the first $T_{opt} = 0.2T$ time steps.
>
> More specifically, at time step $ t $, the latent variable $ z_t $ and the text embedding $ \mathcal{C} $ are fed into the diffusion model. After computing the loss, the gradient is back propagated to update $ \mathcal{C} $. The latent variable $ z_t $ and the updated text embedding $ \mathcal{C} $ are then fed into the diffusion model again to predict the noise and obtain $ z_{t-1} $.
>
> **Q1: Complex prompts generation**
>
> As we have demonstrated in the General Response 2, our method ToMe is not limited to only simple cases. In the main paper Table 2, we show experiments on the T2I-CompBench benchmark, where there are long captions with various complex prompts. There are also complex prompts T2I generations in Fig.12 in the Appendix.
>
> To further show that our method ToMe can be applied to the T2I generation with complex prompts. We adopt the suggestions from the reviewer. That is to achieve T2I generation with prompts including multiple objects or backgrounds with multiple attributes. The first row of T2I generation results shown in Fig.13 (in the rebuttal file) demonstrate the effectiveness of our method ToMe under complex scenarios, where more objects and background with various properties are demanded.
>
>
> **L1: Generalizability**
>
> As we discuss in the General Response 4, we agree with you on this point. T2I models are facing generation limitations for various specific cases. We aim, as a future goal, to generalize our method to various generative models to counter this issue. Current T2I generation models are built on various language models and vision-language models, such as CLIP and T5, and incorporate diverse architectures, including UNet, GAN, and DiT. We are also curious to see if the additivity property truly exists across all these cases, and whether this property is a generalizable feature.

---

### Author Rebuttal · Authors · 2024-08-06

We appreciate all reviewers (**R1**=**jwvK**, **R2**=**mo93**, **R3**=**85Q2**, **R4**=**8gpY**) for their positive feedbacks. They note that this paper is well-written (**R1,R2**); the idea is simple and straightforward (**R1, R2, R4**); that we present interesting analysis over the token properties (**R1, R3**); that we provide extensive experiments over various benchmarks (**R1, R3**); and we well address the object binding in this work (**R4**). Below we respond to general questions raised by reviewers.  We use **W** to abbreviate Weaknesses, **Q** to represent Questions and **L** for Limitations. Note that we utilize the numerical references to cite sources within the main paper.


**General Response 1: User study (R3-W3, R4-W3)**

In this paper, we use the ImageReward [72] model to evaluate human preference scores, which comprehensively measures image quality and prompt alignment. However, since quantitative results may not fully capture these aspects, we now also conduct a user study with 20 participants to enrich the evaluation. Here we compare our method ToMe with SDXL[51], SynGen[56], Ranni[19] and ELLA[28]. Each participant is questioned with 100 images, each generated from these five methods and twenty prompts.

The results are presented in Fig.16 of the rebuttal file. Similar to our proposed GPT-4o benchmark (detailed in Appendix C.5 and Fig.10), we ask the participants (instead of GPT-4o model as we used) to rate the semantic binding into 4 levels and calculate the distribution of each comparison method over these four diverse levels. We can observe that our method better achieve the semantic binding performance by mainly distribute in the highest level 1, while the other methods struggle to obtain user satisfactory results.

**General Response 2: Complex prompts generation (R1-Q1, R3-W1&Q1&L1, R4-W4&Q4)**

Actually, our method ToMe is not limited to only simple cases. In the main paper Table 2, we show experiments on the T2I-CompBench benchmark, where there are long captions with various prompt templates. There are also complex prompts T2I generations in Fig.12 in the Appendix.

To further show that our method ToMe can be applied to the T2I generation with complex prompts. We adopt the suggestions from the reviews. That includes *(1) R1 (jwvK) and R3 (85Q2)*: multiple objects or backgrounds with multiple attributes; *(2) R4 (8gpY)*: long captions with multiple sentences. The generation results are shown in Fig.13 in the rebuttal file.

For the second case, we apply the ToMe method directly, as shown in the last example of Fig.13.
For the first case, we use our method ToMe in an iterative updating way. As an instance, we generate the fourth image example in Fig.13 (left down) with the prompt “a blue cat wearing sunglasses and a yellow dog wearing a pink hat”. To apply ToMe, we merge ‘yellow dog’ into token1 (dog*) and ‘pink hat’ into token2 (hat*). Following that, we merge token1 and token2 into token3. The noun phrases for computing the semantic binding loss corresponding to these three tokens are ‘a yellow dog’, ‘a pink hat’ and ‘a dog wearing a hat’, respectively. Then we update token1 and token2 by our losses proposed in Section 3.2.2.

**General Response 3: Time cost (R2-Q3, R3-W2)**

In Appendix C.4, we present a time cost comparison in Table 3, where we compare performance using 50 inference steps using the SDXL model with float32 version on an A40 GPU.  We demonstrate that our method does not significantly increase inference time while improving semantic binding performance. We further extend this analysis by measuring the time cost with 20 inference steps and various ToMe configurations, as shown in the Table below. We report the time cost (by seconds) along with BLIP-VQA scores across the color, texture, and shape attribute binding subsets. From this table, we can observe that using the token merging (ToMe) technique and entropy loss (Config.C in Table 2), our method achieves excellent performance with minimal additional time cost. Additionally, even with only 20 inference steps, our method, ToMe, maintains high performance with very little degradation.

| method | inference steps | Time Cost | Color | Texture | Shape |
| :--------: | :--------: |:--------: |:--------: |:--------: |:--------: |
SDXL                    |  20        |  18s  |  0.6136 |  0.5449  |  0.5260 |
*ToMe (Config C)*  |   20    |   23s   |   *0.7419*  |  *0.6581*  |   *0.5742* |
**ToMe (Ours)**         |   20    |   45s   |   **0.7612**  |   **0.6653**   |  **0.5974**  |
Ranni (SDXL)     |  50    |  87s    |  0.6893   |  0.6325   |  0.4934 |
ELLA (SDXL)     |  50    |  51s    |  0.7260   |  0.6686   |  0.5634 |
SynGen (SDXL)   |  50    |  67s    |  0.7010   |  0.6044   |  0.5069 |
SDXL                    |  50    |  42s    |  0.6369   |  0.5637   |  0.5408 |
*ToMe (Config C)* |   50    |   56s   |   *0.7525*  |  *0.6775*  |   *0.5797* |
**ToMe (Ours)**         |   50    |   83s   |   **0.7656**  |   **0.6894**   |   **0.6051**  |

**General Response 4: Generalizability to other T2I models (R1-L1, R4-Q2)**

We agree with you on this point. T2I models are facing generation limitations for various specific cases. We aim, as a future goal, to generalize our method to various generative models to counter this issue. Current T2I generation models are built on various language models and vision-language models, such as CLIP and T5, and incorporate diverse architectures, including UNet, GAN, and DiT. We are also curious to see if the additivity property truly exists across all these cases, and whether this property is a generalizable feature.

---

### Decision · Program_Chairs · 2024-09-25

**Decision:**

Accept (poster)

**Comment:**

This paper introduces a novel approach on semantic binding during T2I, this is to associate a given object with its attribute or with other related objects. The approach uses token merging to aggregate relevant tokens into a single token. All reviewers feel positive about the paper. Reviewers think the idea is straightforward and effective, paper is well written with extensive experiments comparing benchmarks. There are a few interesting questions raised by reviewers on complex prompt and importance of entropy loss, authors have provided answers to clarify. The AC agree with all the reviewers and recommend to accept this paper. The AC encourages the authors to add these clarification to the final paper.